# Label-Free Detection of CA19-9 Using a BSA/Graphene-Based Antifouling Electrochemical Immunosensor

**DOI:** 10.3390/s23249693

**Published:** 2023-12-08

**Authors:** Wei Chen, Miaomiao Chi, Miaomiao Wang, Yage Liu, Shu Kong, Liping Du, Jian Wang, Chunsheng Wu

**Affiliations:** Institute of Medical Engineering, Department of Biophysics, School of Basic Medical Sciences, Health Science Center, Xi’an Jiaotong University, Xi’an 710061, China; weiwcchen@xjtu.edu.cn (W.C.); chimiaomiao01@stu.pku.edu.cn (M.C.); wmm15029418463@stu.xjtu.edu.cn (M.W.); liuyage@stu.xjtu.edu.cn (Y.L.); kongshu@mail.xjtu.edu.cn (S.K.)

**Keywords:** electrochemical immunosensor, electrochemical impedance spectroscopy, CA19-9, graphene, antifouling

## Abstract

Evaluating the levels of the biomarker carbohydrate antigen 19-9 (CA19-9) is crucial in early cancer diagnosis and prognosis assessment. In this study, an antifouling electrochemical immunosensor was developed for the label-free detection of CA19-9, in which bovine serum albumin (BSA) and graphene were cross-linked with the aid of glutaraldehyde to form a 3D conductive porous network on the surface of an electrode. The electrochemical immunosensor was characterized through the use of transmission electron microscopy (TEM), scanning electron microscopy (SEM), atomic force microscope (AFM), UV spectroscopy, and electrochemical methods. The level of CA19-9 was determined through the use of label-free electrochemical impedance spectroscopy (EIS) measurements. The electron transfer at the interface of the electrode was well preserved in human serum samples, demonstrating that this electrochemical immunosensor has excellent antifouling performance. CA19-9 could be detected in a wide range from 13.5 U/mL to 1000 U/mL, with a detection limit of 13.5 U/mL in human serum samples. This immunosensor also exhibited good selectivity and stability. The detection results of this immunosensor were further validated and compared using an enzyme-linked immunosorbent assay (ELISA). All the results confirmed that this immunosensor has a good sensing performance in terms of CA19-9, suggesting its promising application prospects in clinical applications.

## 1. Introduction

Carbohydrate antigen 19-9 (CA19-9) is a kind of serum Lewis (a) carbohydrate antigen, and it is one of the best-validated biomarkers in pancreatic cancer, with 80% sensitivity [1]. The level of CA19-9 may also be elevated in patients who suffer from other gastrointestinal malignancies, such as colon cancers, liver cancers, or gastric cancers, while it is usually lower than 37 U/mL in healthy people. Therefore, the level of CA19-9 can be used to evaluate the cancer stage and to predict long-term survival. In addition, it is much more effective to combine CA19-9 with other cancer biomarkers in the screening of serum samples from cancer patients [2]. Therefore, the development of a rapid, simple, and sensitive method for the detection of CA19-9 has great application prospects.

Currently, the detection of CA19-9 is mainly based on immunoassay-related strategies, including enzyme-linked immunosorbent assays (ELISAs) [3], electrochemical biosensors [4,5,6,7], photoelectrochemical biosensors [8], electrochemiluminescence [9], fluorescent or colorimetric assays [2], and giant magneto-resistance biosensors [10]. Among these various methods, electrochemical biosensors have great potential to be miniaturized and automated for point-of-care applications in clinical diagnostics [11,12,13], and they can also be cost-effective and portable, with low-power instrumentation. In particular, the electrochemical impedance spectrum (EIS) technique, as a simple label-free analytical method, can be utilized for the development of immunosensors and has great prospects in terms of point-of-care applications for various biomarkers [14].

The complex components in biological samples can result in the biological fouling of electrochemical electrode surfaces via non-specific adsorption and adhesion, which decrease the electrochemical current and the analytical performance of sensors. This is still a major obstacle to the successful commercial development of various electrochemical immunosensors [12,15]. Therefore, the surface treatment of electrochemical electrodes is crucial for the antifouling performance of electrochemical sensors. Currently, antifouling strategies mainly include physical antifouling (such as nanoporous surfaces) and chemical antifouling (such as bovine serum albumin (BSA), antifouling polymer layers, and hydrogel) [15,16,17]. For the physical antifouling strategy, various micro/nanostructures can be engineered on the transducer surface in order to fabricate a size-selective diffusional barrier of large non-specific molecules, which can also permit the diffusion of small analytes to the underlying transducer [4,6,16]. For the chemical antifouling strategy, a wide range of antifouling molecules can be incorporated into electrochemical interfaces via self-assembly, electro-grafting, or polymerization methods. BSA is a common and cheap protein that has been broadly used in biological antifouling due to its abundance and superior biostability in the body. Antifouling polymers contain various chemicals, such as the commonly used polyethylene glycol (PEG), oligoethylene glycol (OEG), poly(vinyl alcohol) (PVA), zwitterionic polymers, and peptides with a mixed charge [18,19,20,21,22]. However, for most antifouling molecules, the formed antifouling layer may hinder electron transfer and decrease the current at the electrochemical interface, resulting in a significant decrease in electrochemical sensing performance [23]. Therefore, the balance of the antifouling and electrochemical performance of interfaces is an important issue. Incorporating antifouling molecules and conducting nanomaterials is an effective strategy to form an antifouling layer with 3D conductive nanocomposites, and it has been demonstrated to be suitable for the development of electrochemical immunosensors with excellent performance [24,25,26].

Here, we proposed an effective BSA/graphene-based antifouling electrochemical immunosensor, in which BSA and graphene were cross-linked with glutaraldehyde (GA) to form a 3D conducting antifouling layer on the surface of an electrochemical electrode. Graphene is a kind of thin, two-dimensional carbonaceous nanomaterial with superior conducting performance. The cross-linking of BSA and graphene with GA can build a stable 3D porous conducting network, which can maintain the antifouling behavior of BSA and the conducting performance of graphene at the same time. In addition, this BSA/graphene-based antifouling film can provide covalent reaction sites for the further effective immobilization of antibodies. By employing electrochemical impedance spectroscopy (EIS), a label-free electrochemical immunosensor was developed for the rapid and sensitive detection of CA19-9 in complex human serum samples. All the obtained results confirmed the good performance of this electrochemical immunosensor for the label-free detection of CA19-9. It has great potential and promising prospects in terms of application for the detection of CA19-9 in clinical samples to aid in the diagnosis of cancer diseases. It is worth noting that the electrodes and antibodies used in this study were only used to demonstrate the technical feasibility of this novel approach to the label-free detection of CA19-9.

## 2. Materials and Methods

### 2.1. Materials and Apparatus

The following were used in this study: graphene (XFNANO, Nanjing China, with a diameter of 0.5 μm–5 μm, thickness of 0.8 nm, and purity of 99%), BSA (Sigma-Aldrich, Shanghai, China, product no. V900933), glutaraldehyde (GA, Sigma-Aldrich, Shanghai, China, no. G7776), N-ethyl-N′-(3-dimethylaminopropyl)carbodiimide (EDC, Sigma-Aldrich, Shanghai, China, no. 03449), N-hydroxysuccinimide (NHS, Sigma-Aldrich, Shanghai, China, no. 130672), 2-(N-morpholino)ethanesulfonic acid (MES, Sigma-Aldrich, Shanghai, China, no. M3671), ethanolamine (MEA, Aladin, Shanghai, China, E103810), anti-CA19-9 antibody (GeneTex, Irvine, CA, USA, GTX635391), CA19-9 protein (Fitzgerald, Louisville, KY, USA, 30-AC15), normal human serum (Solarbio, Beijing, China, SL010), a CA19-9 ELISA test kit (Bio-Swarmp, Beijing, China, HM10541), transmission electron microscopy (TEM) (JEOL JEM-2100Plus, Tokyo, Japan), and scanning electron microscopy (SEM) (TESCAN MAIA3 LMH, Brno, Czech Republic). All electrochemical measurements were performed using an electrochemical workstation (Metrohm Dropsens STAT-I 400, Asturias, Spain).

### 2.2. Preparation of BSA/Graphene Nanocomposites 

BSA/graphene nanocomposites were prepared by mixing 1.0 mg graphene and 15.0 mg BSA into 1 mL phosphate-buffered saline (PBS). The mixture was sonicated under a tip sonicator (125 W and 20 kHz) for 30 min with on/off intervals of 1 s at 50% amplitude, yielding an opaque black solution. After centrifugation at 3500 rounds per minute (rpm) for 15 min, a semi-transparent solution was recovered, and the black precipitate was discarded. The obtained BSA/graphene nanocomposite was stored at 4 °C for further application. The topography of the BSA/graphene nanocomposite was characterized through the use of TEM.

### 2.3. Preparation and Functionalization of BSA/Graphene/GA/Antibody-Modified Electrodes

A gold electrode (AuE, with a diameter of 3 mm) was firstly polished with 1.5 μm and 0.5 μm alumina slurries and rinsed with ultrapure water. The polished gold electrode was chemically cleaned with a freshly prepared piranha solution (volume ratio of H_2_SO_4_ and H_2_O_2_ was 3:1) and then thoroughly rinsed with ultrapure water for further modification.

The BSA/graphene nanocomposite was directly mixed with 70% GA at a volume ratio of 69:1. The well-cleaned gold electrode was immersed into the BSA/graphene/GA solution and maintained in a water-saturated atmosphere for 24 h at room temperature. After that, the modified gold electrode was thoroughly rinsed with PBS in a shaker for 30 min. The BSA/graphene/GA-modified electrodes were further functionalized with antibodies using carbodiimide chemistry. Briefly, the BSA/graphene/GA electrode was incubated with 400 mM EDC and 200 mM NHS in 0.1 M MES buffer (pH 6.0) for 30 min, rinsed with ultrapure water, and dried at room temperature. Then, 20 μg/mL of the anti-CA19-9 antibody was prepared in PBS with 0.5% glycerol. The activated gold electrode was reacted with the anti-CA19-9 antibody solution in a water-saturated atmosphere overnight at 4 °C, and it was rinsed and washed with PBS in a shaker for 30 min. Next, 1 M MEA solution was prepared in PBS and adjusted to pH 7.4 with HCl. The electrode was incubated with 1 M MEA at room temperature for 30 min to quench the unreacted active groups and then further blocked with 1% BSA solution for 1 h at room temperature. Finally, the electrode was thoroughly rinsed in ultrapure water for further application.

The diluted BSA/graphene nanocomposites were characterized using UV spectroscopy before and after the addition of GA to elucidate the conjugation-induced changes in the absorption spectra. The morphology of the BSA/graphene/GA-modified gold electrode was characterized through the use of SEM and an atomic force microscope (AFM). Each functionalization process of the electrodes was electrochemically characterized in a three-electrode electrochemical cell via cyclic voltammetry (CV) and electrochemical impedance spectroscopy (EIS) measurements. The redox aqueous solution in the electrochemical cell was 5 mM K_4_Fe(CN)_6_/K_3_Fe(CN)_6_ with 1 M KCl. The functionalized gold electrode was used as the working electrode, Ag/AgCl was used as the reference electrode, and Pt wire was used as the counter electrode. CV was performed in a voltage range from 0 V to 0.7 V, with a scan rate of 100 Mv s^−1^. EIS was conducted with a 5 mV amplitude at open-circuit potential in a frequency range from 1 MHz to 1 Hz.

### 2.4. Detection of CA19-9 in PBS and Human Serum

To detect CA19-9 proteins using the well-prepared electrochemical immunosensors, standard CA19-9 proteins were diluted with PBS or normal human serum with a series of concentrations (6.25 U/mL, 12.5 U/mL, 25 U/mL, 50 U/mL, 100 U/mL, 200 U/mL, 300 U/mL, 500 U/mL, and 1000 U/mL). The target CA19-9 solution was incubated on the BSA/graphene/GA/antibody-modified electrodes for 1 h. After being rinsed with PBS solution, the electrode was measured for EIS in 5 mM K_4_Fe(CN)_6_/K_3_Fe(CN)_6_ with 1 M KCl. The electrochemical impedance was fitted with a modified Randles equivalent circuit model. CA19-9 (25 U/mL and 50 U/mL in human serum) was also tested with an ELISA kit according to the instructions.

### 2.5. Statistical Analysis

All the electrochemical measurements were performed at least 3 times. The data are shown as mean ± standard deviation (SD). The limit of detection was calculated as the corresponding concentration value of the calibration curve according to the principle of 3δ.

## 3. Results

### 3.1. Working Principle of the Antifouling Electrochemical Immunosensor 

To enhance the performance of the electrochemical immunosensor for detecting the protein biomarker in complex clinical samples, we proposed an antifouling label-free electrochemical impedance biosensor based on a 3D porous BSA/graphene/GA matrix (Figure 1). The BSA and graphene nanocomposites were cross-linked via GA to modify the surface of the gold electrode and form a conducting antifouling interface, where BSA acted as a major component of the antifouling effect, and the conducting 2D nanomaterial (graphene) was used to sustain the electron transfer of the electrode. The antibody of the target protein can be covalently immobilized onto the layer of BSA/graphene/GA with the aid of EDC and NHS. After being further blocked with MEA and BSA, this modified electrochemical electrode can be employed to capture the target protein via specific immune recognition between antibodies and antigens. The specific capture of target proteins will hinder the electron transfer at the interface from the electrolyte to the electrode, which can be monitored via an electrochemical impedance spectrum in a label-free manner. 

### 3.2. Characterization of BSA/Graphene-Modified Immunosensor 

In general, it is difficult to disperse graphene powder in PBS directly. However, when graphene is dispersed in a BSA solution, BSA proteins can adsorb onto the surface of the graphene sheet through hydrophobic and electrostatic interactions, resulting in the formation of a homogeneous dispersion of the BSA/graphene nanocomposite. During the preparation of the BSA/graphene nanocomposite solution, the concentration of BSA in PBS was optimized from 1 mg/mL to 15 mg/mL. It was found that the nanocomposite solution became more stable and homogeneous only when the concentration of BSA was up to 10 mg/mL (Appendix A). TEM was performed to better reveal the topography of the BSA/graphene nanocomposite. The two-dimensional single-layered and stacked graphene can be observed in Figure 1a. After the BSA/graphene nanocomposites were cross-linked with GA, a 3D sponge-like conducting protein matrix was generated. This was confirmed via an SEM analysis (Figure 1b), revealing that a relative densely packed nanocomposite film formed on the surface of the gold electrode. The GA-assisted cross-linking reaction produced 3D molecular networks of BSA/graphene. The detailed mechanism has been well discussed in a previous report [25]. In brief, GA can react quickly to yield polymers of pyridine in the presence of the primary amines of BSA, resulting in the rapid formation of structural 3D glue molecular networks. This reaction can also increase the UV absorbance at 265–270 nm (Figure 1c). In addition, the surface morphology of the modified electrodes was further characterized with atomic force microscopy (AFM), and the result showed that the roughness of the gold electrode after the modification of the BSA/graphene/GA nanocomposites was $R_{a}=2.43\pm 0.15nm$ (Figure 1d). All the results confirmed that the BSA/graphene nanocomposite formed a stable 3D porous molecular network in the presence of GA on the surface of the electrode.

The electrochemical performance of different BSA/graphene/GA-modified gold electrodes was evaluated for the BSA and graphene nanocomposite at different ratios. Considering that the nanocomposite solution was less stable when the concentration of BSA was less than 10 mg/mL, here, we only tested the concentrations of 10 mg/mL and 15 mg/mL (named as BSA10 and BSA15). The final concentrations of graphene and GA were kept at 1 mg/mL and 1.4% (*v*/*v*), respectively. To assess the overall electrochemical quality and state of the solid–liquid interface, the potential separation between the oxidation peak and the reduction peak (ΔEp) and current changes were calculated according to cyclic voltammetry. Typical cyclic voltammetry showed that different modifications resulted in obviously different electron transfer kinetics (Figure 2a). The gold electrodes showed partial passivation after different modifications due to BSA hindering the electron transfer. ΔEp and current changes were calculated and compared between the groups of BSA10 and BSA15. The statistical results (Appendix A) showed that BSA15/graphene1/GA displayed a relatively low ΔEp (102 ± 16 mV) and a high current (90 ± 6%), indicating that this BSA15/graphene1/GA coating maintained good electron transfer characteristics and exhibited better performance than the other coating protocols. Voltammograms at different scan rates were also tested and compared with those of the bare gold electrode in order to evaluate the mass transport process on the BSA15/graphene1/GA-modified electrode (Figure 2b,d). It was found that the peak current was linear to the square root of the scan rate with the increase in the scan rates in both the bare electrode and the BSA15/graphene1/GA-modified electrode (Figure 2c), indicating a diffusion-limited process of electroactive species at the interface. All the results demonstrate that the incorporation of conducting graphene into the BSA network improved the overall migration of the electroactive species at the interface of the electrode and electrolyte solution, which can be attributed to the formation of a porous coating membrane with the GA-assisted cross-linking of BSA and nanomaterials [24,25].

During the process of sensor preparation, the electrochemical characteristics of the electrodes were monitored step by step using CV and EIS (Figure 3). The voltammograms and Nyquist diagrams changed substantially after the covalent immobilization of the CA19-9 antibody and the BSA blocking. Figure 3a shows clear redox peaks on the bare gold electrode and BSA15/graphene/GA-modified electrode. Figure 3b shows that the electrochemical impedance decreased slightly after the modification of BSA15/graphene/GA. After the CA19-9 antibody was linked covalently on the BSA15/graphene/GA-modified electrode, the formation of a protein layer at the interface of the electrode and electrolyte resulted in a dramatic decrease in redox peak currents and an increase in impedance, which confirmed the successful immobilization of the CA19-9 antibody on the BSA15/graphene/GA electrode. In addition, ethanolamine and BSA were incubated onto the electrode surface to block the unreacted active sites, resulting in a further slight decrease in both the correlated redox currents and impedance. Additionally, the surface morphologies of the modified electrodes were also characterized with AFM after the capture of the CA19-9 antibody and antigen; the results are shown in Appendix A. It was found that the capture of the antibody and antigen onto the electrode did not result in significant changes in the surface roughness, which indicates that only the surface roughness of AFM cannot characterize the capture of a single layer of protein molecules. The electrochemical sensor was well characterized using CV and EIS, and it is suitable for the further detection of the target protein CA19-9 via specific binding with the antibody. 

### 3.3. Performance of Electrochemical Immunosensor for the Detection of CA19-9

To examine the performance of the developed electrochemical immunosensor, different concentrations of the CA19-9 protein were diluted with a standard PBS buffer or human serum. The responses of the electrochemical immunosensor were recorded by monitoring the changes in electrochemical impedance (Figure 4). The results demonstrate that the electrochemical impedance increased with the increase in the target CA19-9 concentrations, and the immunosensor exhibited similar responses regardless of whether the dilution buffer was the PBS buffer or human serum (Figure 4a,b). The responsive results were quantified by fitting the electrochemical impedance data with a modified Randles circuit model (R_s_(CPE(R_ct_Z_w_))) (Appendix A), in which R_s_ represents the resistance of the solution; the constant phase element (CPE) corresponds to a capacitor of the electrochemical interface with a constant phase; the charge transfer resistance (R_ct_) represents the difficulty of electron transfer at the electrode interface and corresponds to the diameter of a high-frequency semicircle in the Nyquist plot; and the Warburg resistance (Z_w_) represents the diffusion process of redox probes from the electrolyte to the electrode surface [27]. The specific capture of target antigen proteins mainly resulted in changes in R_ct_ by interfering with the electrode/electrolyte interface. Therefore, the relative change in R_ct_ was used to quantity the target protein-induced impedance changes, which was defined as ΔR_ct_ = R_ct_ (after target capture)-Rct (after blocking). The concentration-dependent calibration curves are shown in Figure 4c,d. The detection range was from 6.25 U/mL to 1000 U/mL. The detection limit of CA19-9 was calculated to be as low as 13.5 U/mL in serum according to the principle of S/N = 3.

Due to the complex components in human serum, the selectivity of this electrochemical biosensor was evaluated by comparing the results of the response to CA19-9 using the same concentrations of the PBS buffer and normal human serum. Figure 4c,d and Figure 5a show the comparable responses in both solution backgrounds, indicating that this electrochemical immunosensor had good selectivity to the target protein CA19-9. When the BSA15/graphene/GA-modified electrodes were kept in PBS, 1% BSA, and human serum for 2 days, the currents remained comparable to the original states (Figure 5b, 98.9% in PBS; 91.9% in 1% BSA; 83.8% in serum). The BSA15/graphene/GA antibody-modified electrodes showed good selectivity towards CA19-9, even though there are various other background proteins with different concentrations in normal human serum, which could be attributed to the good antifouling performance of the BSA15/graphene/GA modification. The GA-assisted cross-linking between BSA and graphene formed a homogeneous 3D conductive coating film, which balanced the BSA-based antifouling and graphene-assisted electron transfer. All the results prove that this kind of surface modification had good antifouling performance and reduced the non-specific binding in the complex clinical samples.

The stability of the modified electrodes was also evaluated at 4 °C for 10 days either in a dry state or in a PBS solution by testing the current changes and electrochemical impedance changes. The results show that the BSA15/graphene/GA-modified electrodes demonstrated good stability under both conditions, and the R_ct_ remained stable for 10 days (Figure 5c, 104.4% in dry; 96.4% in PBS), indicating that these BSA15/graphene/GA-modified electrodes exhibited good stability performance. Furthermore, the reproducibility performance was tested with three independent BSA/graphene/GA-modified electrodes. The relative standard deviation (RSD) was 8.6%, indicating the acceptable reproducibility and the possibility of developing disposable immunosensors.

In addition, the performance of this electrochemical immunosensor was further validated using an ELISA kit. Standard CA19-9 was spiked into normal human serum at the concentrations of 25 U/mL and 50 U/mL, which were tested with the electrochemical immunosensor and the ELISA kit, respectively. The results (Figure 5d) show that the detected value of the electrochemical immunosensor matched well with that of ELISA. All the results indicate that this electrochemical immunosensor has good sensing performance and great potential for the development of disposable biosensors for real clinical application. 

Several electrochemical immunosensors have been reported for the detection of CA19-9. For most electrochemical immunosensors, the sensitive material (antibody) is usually immobilized onto the electrode surface covalently [21]. Various nanomaterials have been employed to enhance the sensing performance in combination with corresponding sensing techniques [28,29,30]. The performance of the immunosensor in this paper is comparable to that of previously reported biosensors (Appendix A). In addition, considering that the level of CA19-9 was elevated significantly, the large detection range is one of the main advantages of the electrochemical immunosensor developed here, which can ensure the quantification of CA19-9 in various patients. The cross-linking of BSA and graphene with GA ensures good antifouling performance in the detection of serum samples.

## 4. Conclusions

A label-free antifouling electrochemical immunosensor was successfully developed by cross-linking BSA and graphene with the aid of GA. The obtained nanocomposites formed a stable porous 3D conductive antifouling layer on the surface of an electrode, which maintained both the good electron transfer and antifouling performance at the electrochemical interface at the same time. After the antibody was further immobilized covalently onto the BSA/graphene/GA film, the target protein CA19-9 could be quantified directly via the electrochemical impedance spectroscopy technique in a label-free manner. This electrochemical immunosensor exhibited good sensing performance towards CA19-9 proteins in both a buffer and human serum. The responsive range was from 13.5 U/mL to 1000 U/mL, with a detection limit of 13.5 U/mL. This immunosensor also showed good selectivity and stability in complex serum samples. The detection result was also validated using an ELISA kit. All the results demonstrate that incorporating antifouling BSA and conductive graphene is a simple and effective strategy to build an electrochemical interface. This label-free electrochemical immunosensor provides a rapid and effective method for the evaluation of protein biomarkers in complex samples, which has great application prospects in cancer diagnosis. With the aid of microfabrication techniques, this electrochemical immunosensor can be further incorporated into portable sensing devices for point-of-care applications.

## Data Availability

The data presented in this study are available on request from the corresponding author. The data are not publicly available due to privacy restrictions.

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
