# Peer review of "Label-Free Detection of CA19-9 Using a BSA/Graphene-Based Antifouling Electrochemical Immunosensor"

_sensors, 2023, doi:10.3390/s23249693_

Round 1

Reviewer 1 Report

Comments and Suggestions for Authors

The article titled “Label-free detection of CA19-9 using a BSA/Graphene-based 2 antifouling electrochemical immunosensor' presents the development of the detection process of CA19-9 by crosslinking bovine serum albumin (BSA) and graphene.

The following points need to be addressed before the possibility of publication of this work:

1-      Moderation of the English language is really required.

2-      The full name of CA19-9 should be written in the abstract rather than its abbreviation.

3-      Add a table to compare your findings with other techniques regards the described detection process.

4-      Support the details of Figure S2 with references.

5-      You should add some data of your characteristics TEM,  UV spectroscopy and AFM after the immobilisation process of antibody and antigen.

6-      Add optical images and XPS data of synthesized BSA/Graphene nanocomposite to further confirm their structure and quality.

Comments on the Quality of English Language

English language should be improved as there are many mistakes in grammar.  

Reviewer 2 Report

Comments and Suggestions for Authors

This work proposed an antifouling electrochemical impedance spectroscopy (EIS) immunosensor for label-free detection of CA19-9, a cancer biomarker. The glutaraldehyde-crosslinking of bovine serum albumin (BSA) and graphene was performed to form a 3D conductive porous network on the surface of electrode. This electrochemical immunosensor showed excellent antifouling performance to detection CA19-9 in human serum samples, with comparable results with ELISA. This manuscript is well organized. But some revisions should be considered before recommendation for publication:

1) As to Scheme 1, actually, the immobilized antibody may be randomly distributed, with many binding sites (Fab) buried, which results in a low capture efficiency. Have the authors considered the optimal density of antibody on the electrodes to guarantee the maximum exposure of binding sites to antigens in samples?

2) Section 2.2, as to the preparation of BSA/Graphene nanocomposites, the authors claimed that the feeding ratio has been optimized so that the nanocoposite becomes more stable and homogeneous. How to confirm this conclusion? The data should be added. Moreover, excess BSA may affect the electrochemical performances of graphene matrix. What about the effect of BSA in this work?

3) Fig. 2c, the linear relationship coefficient for each curve can be added in the figure.

4) Figure 5, to demonstrate that the immunosensor exhibits good selectivity and stability, more data should be provided, such as interferences from other common biomarkers to avoid cross-talk.

5) How to regenerate the electrodes? What about the electrochemical performances of modified electrodes after regeneration?

Round 2

Reviewer 1 Report

Comments and Suggestions for Authors

None.

Reviewer 2 Report

Comments and Suggestions for Authors

The authors have addressed my issues. I've no further comments now.